# Assessing Students' Awareness of 4Cs Skills after Mobile-Technology-Supported Inquiry-Based Learning

**Manolis Kousloglou \*** , **Eleni Petridou, Anastasios Molohidis and Euripides Hatzikraniotis**

Physics Department, Aristotle University of Thessaloniki, 54124 Thessaloniki, Greece;
elepet@physics.auth.gr (E.P.); tasosmol@physics.auth.gr (A.M.); evris@physics.auth.gr (E.H.)
\* Correspondence: ekouslog@physics.auth.gr

**Abstract:** Inquiry-based learning (IBL) is essential to Science Education since it improves students' conceptual comprehension, higher-order thinking abilities, and interpersonal skills. Mobile technology (mIBL) promotes active learning, facilitates access to learning materials, and enhances IBL in terms of mobility and rapid feedback. This study assesses the 9th grade students' awareness of 4Cs skills (Collaboration, Communication, Critical thinking and problem solving, and Creativity) after participating in mobile-technology-supported inquiry-based Teaching Learning Sequence (TLS). Additionally, the study investigates the qualities/characteristics students cite before and after the TLS in relation to these skills. The results of a questionnaire administered to students indicate that their awareness of these skills has increased as a result of the TLS. Specifically, the TLS seems to have a greater impact on Collaboration and Communication than on Critical thinking and problem solving, and Creativity. An additional qualitative investigation of students' written answers to explanatory open-ended questions before and after the entire procedure found intriguing data confirming their shift in 4Cs awareness.

**Keywords:** inquiry-based learning; mobile learning; MBL; sensors; mIBL; 21st century skills; Science Education

## 1. Introduction

Due to the trend of globalisation, several competencies are seen as essential for enhancing the technological literacy and competitiveness of individuals. According to reports, young people demand transferable abilities as opposed to technical skills, which are regarded as specialised to a certain activity, career, or industry [1,2]. The 21st century frameworks provide methods to define the abilities that students must acquire to enter the future workforce; consequently, it is the responsibility of educators to determine if the present competencies and learning methods are suited to achieve this [3].

Several organisations, such as the "P21 Framework" and the "Society for Technology in Education", have identified different types of skills that students should master in the 21st century [3–5]. Collaboration, Communication, Critical thinking and problem solving, and Creativity are considered to be of utmost importance. Students should acquire these skills for three primary reasons: (a) these skills are difficult to teach and assess, thus they are seldom included in the curriculum; (b) these skills are essential for all students in the era of globalisation; and (c) these skills are essential for any career [6].

Inquiry-based learning (IBL) increases students' conceptual comprehension, critical thinking, problem solving, and teamwork abilities, according to several research studies [7,8]. Mobile-technology-supported inquiry-based learning (mIBL) is a popular teaching strategy in secondary Science Education because digital mobile technology boosts learning in inquiry-based situations [9–11]. However, there is a shortage of evidence regarding mIBL in secondary Science Education due to the studies' techno-centric emphasis [12]. Furthermore, there is a gap in the research about the relation between mIBL and the development of 21st century skills in Science Education.

Our research assesses secondary school students' awareness of 4Cs skills through mobile-technology-supported inquiry-based learning. The significance of this paper is its contribution to fill the need for further evidence-based insights into the relationship between mIBL and the development of students' high-order thinking skills. The structure of the paper includes an introduction to 4Cs skills and mobile-technology-supported inquiry-based learning, the presentation of the methods applied in the research, the results, and the conclusions.

## 2. Background

This section provides a brief theoretical framework for 21st century skills and analyses the learning skills, as stated by the P21 Framework. The inquiry-based learning (IBL) strategy is then described, along with the definitions of microcomputer-based labs (MBL) and mobile learning (m-learning), as well as their combination, i.e., mobile-technology-supported inquiry-based learning (mIBL), of which was the primary instructional technique utilized during our interventions. Finally, the relationship between mIBL and 4Cs skills is mentioned.

### 2.1. The 21st Century Skills

There are several perspectives on the precise substance and meaning of 21st century skills, which all highlight what students can accomplish with information and how they use what they learn in actual circumstances [4,5]. Society for Technology in Education (ISTE) acknowledged that in a world that is becoming more digital, students require abilities in the following areas: (1) collaboration and communication; (2) critical thinking, problem solving, and decision making; (3) creativity and innovation; (4) research and information fluency; (5) digital citizenship; and (6) operations and concepts of technology. The Framework for 21st Century Learning [13] was created by the P21 Framework, a collaborative organisation that generated a cohesive, common vision for learning [3]. The P21 Framework outlines three types of competencies: (1) learning skills (collaboration and communication, critical thinking, problem solving, creativity and innovation); (2) literacy skills (information literacy, media literacy, and ICT literacy); and (3) life skills (flexibility and adaptability, initiative and self-direction, social and intercultural skills, productivity and accountability, and leadership and responsibility) [3]. Our study focuses on learning skills as described by the P21 framework.

Collaborative learning is a kind of learner-to-learner contact that involves learning, sharing of authority and acceptance of responsibility, and respect for the skills and contributions of each group member. Along with assertiveness, responsibility, and empathy, collaboration is often seen as a social skill. Individual efforts must give way to teamwork, and autonomy must give space to community, if crucial concerns are to be effectively addressed [5,14,15]. Collaboration skills are necessary for students to engage and compete in the 21st century because they enable them to communicate with their peers and share their thoughts and ideas for accomplishing learning objectives and promote the community. Based on the above, the students' views on acknowledging group members' abilities through teamwork, accomplishing the learning objectives, promoting community, sharing of authority and developing assertiveness will be examined.

Communication is the ability to express thoughts and ideas effectively using oral, written, and nonverbal abilities in various formats and circumstances. Communication is motivated and directed by the desire to attain certain goals and is supported by perceptual, cognitive, emotional, and behavioural activities [2,16]. Effective communication skills encompass characteristics such as empathy, understanding, and active listening, respect for a person's dignity, integrity, and autonomy as well as the capacity to explore and share ideas and thoughts in a warm, non-judgmental, and pleasant manner. Therefore, the students' views on participating in dialogue for expressing thoughts and ideas, encouraging peers' effort, communicating to attain certain goals, developing empathy or respect for peers' personality, and developing abilities of active listening will be examined.

Critical thinking as a skill refers to the capacity to evaluate the credibility of a claim or piece of information and reach a judgement regarding what to believe or do. It is also a cognitive tool that students employ to evaluate their methods and beliefs in a reflective manner [2,5]. Students who can monitor and analyse their own cognitive processes are more likely to display high-quality thinking as a result of metacognition (or thinking about thinking). When students think critically, they evaluate the outputs of their thought processes, such as the quality of a choice or the efficacy of a problem solution. Several other essential student learning outcomes, such as metacognition, motivation, collaboration, and creativity, are related to critical thinking abilities [17,18]. Problem solving refers to the ability to detect problems, acquire and assess relevant information, propose viable solutions, and select the most effective technique for addressing the problem [2]. The OECD defines problem solving in the context of education as follows: 'The capacity of students to understand problems situated in novel and cross-curricular settings, to identify relevant information or constraints, to represent possible alternatives or solution paths, to develop solution strategies, and to solve problems and communicate the solutions' [19] (p. 3). Ideally, such problems should be authentic and founded in everyday situations [5]. Based on the above, the students' views on reflecting/monitoring, evaluating processes of thought, proposing alternative and viable solutions, and detecting and understanding the problem will be examined.

Although there is no consensus among scholars about the definition of creativity, the majority think that it is a path that students enjoy [20,21], and involves the production of anything regarded as innovative or beneficial in a specific social context. The researchers concur that the final result of creativity might be a physical object, or an idea generated by developing, refining, analysing, and assessing external stimuli and data [2,5]. An intriguing definition of creativity is provided by Walia [22] (p. 242): "Creativity is an act that arises from a perception of the environment that acknowledges a certain disequilibrium, resulting in productive activity that challenges patterned thought processes and norms, and gives rise to something new in the form of a physical object or even a mental or emotional construct". Therefore, the students' views on producing innovation (tangible or intangible), producing something beneficial in a social context, and gaining pleasure from producing innovation will be examined.

### 2.2. Mobile-Technology-Supported Inquiry-Based Learning (mIBL) in Science Education and 21st Century Skills

Inquiry-based learning (IBL) is a learning approach in which students assume the role of scientists, as they develop questions, formulate appropriate hypotheses and design activities and experiments to test them, they analyse, understand, and explain the results of their experiments, they draw conclusions, and communicate their findings. Therefore, they undertake investigations to generate new knowledge based on the gathered data [8,12,23]. The inquiry process includes references to communication skills, planning, and evidence selection, all of which enhance thinking and problem solving [6]. Numerous studies recommend IBL as a crucial element in Science Pedagogy because it improves students' conceptual comprehension, critical thinking, problem solving and collaboration skills. IBL has the potential to engage students in a real scientific discovery process by giving them a sense of classroom learning achievement and making learning more enjoyable [7,8,24].

Mobile learning refers to the use of mobile devices to aid learning, such as smartphones, laptops, tablets, and wireless sensors. Numerous concepts are associated with m-learning, including learning in multiple contexts and through social interactions; ubiquitous learning (learning anywhere and at any time); context-aware ubiquitous learning, which emphasises the support of learning across contexts; authentic learning, which focuses on real-world problems to create an attractive learning environment; customization of access to information in order to develop new skills; and student-centred learning [7,11,12,25–27]. Science is built on the exploration of the physical world, and digital mobile technology is deemed suitable for supporting this research because it provides the means to make it more accessible

and pervasive [9,10]. Many researchers have also reported that adopting effective learning methods in mobile learning activities might be an empowering strategy for fostering students' 21st century skills [2].

Many researchers mention the contribution of Science Education to the development of 4Cs skills. Authors concur that during science learning, students must build their collaboration abilities since they must be able to work with teammates to solve problems [2,14,28]. Moreover, Communication is also considered a vital component of education as it has a significant impact on students' cognitive perception [29,30]. There are no disagreements on the significance and prevalence of Critical thinking and problem solving in Science Education across various educational systems. Problem solving or locating suitable solutions to challenges is one manner in which Critical thinking and Science are related [31]. The link between Critical thinking and Science Education includes practices, skills, and processes such as problem-finding/identification and obtaining information. Several countries have attempted to include Critical thinking in Science Education, realising that living in a diverse society requires citizenship ability [31,32].

Therefore, the literature suggests that the 4Cs skills are essential components of Science Education and can be enhanced by inquiry-based learning (IBL) as well as by using portable digital devices (m-learning). Mobile-technology-supported inquiry-based learning (mIBL) intends to use mobile technologies to support the inquiry process and drive students to create and share their knowledge, and mIBL enhances IBL in terms of mobility and feedback speed and has a beneficial effect on student–teacher interaction [33,34]. According to relevant research, inquiry-based learning strengthens students' reasoning and thinking skills through inquiry activities, and mobile learning activities might improve critical thinking by encouraging collaborative learning [35,36].

### 2.3. Research Questions

The purpose of our study is to assess secondary school students' awareness of 4Cs skills after participating in mobile-technology-supported inquiry-based learning. Specifically, the research questions of the study are:

- To what extent does mIBL affect students' awareness about Collaboration, Communication, Critical thinking and problem solving, and Creativity skills?
- What qualities/characteristics do students cite before and after the TLS in relation to Collaboration, Communication, Critical thinking and problem solving, and Creativity skills?

## 3. Materials and Methods

### 3.1. The Sample and the Context

This study took place during the second semester of the 2021–22 school year, in a high school in Kavala, N. Greece. The sample consisted of ten ninth graders (15 years old) who participated voluntarily in the school's science club; school clubs operate weekly, after the end of the school schedule. This club was founded by one of the authors of this paper, who was also the students' science teacher. Four females and six males, who earned high grades in Physics, were engaged in a mobile-technology-supported inquiry-based learning (m-IBL) Teaching Learning Sequence (TLS). The students had no experience with inquiry-based learning and had never engaged in mobile learning activities, while being proficient with the use of their smart phones.

### 3.2. The Research Tools

A questionnaire, designed by Hwang et al. [2], with a 3-point Likert scale ranging from "disagree" (value: 1) to "agree" (value: 3), was used in our study to evaluate the students' awareness of 4Cs skills. A 3-point Likert scale was selected because of the small sample. The questionnaire includes twenty-three question items, which are grouped into the skills categories, namely, Collaboration, Communication, Critical thinking, problem solving and Creativity, with 5, 6, 4, 4, and 4 items, respectively. The questionnaire was independently

translated into the Greek language by a team of three translators/experts who were familiar with both the subject concept and the literature on the adaptation of a questionnaire. The translators were gathered to compare and discuss any discrepancies between the translated versions. It is significant to highlight that in the context of the validation and adaptation of the questionnaire to Greek culture, prior to the TLS, the questionnaire was given to ninth-grade students in order to make sure that the students understand the meaning of all the items of the questionnaire, ensuring that their replies are accurate and comparable (before and after the TLS). So, ten ninth-grade students completed the Greek version of the questionnaire, and a discussion was facilitated between the students and the teacher to assess their understanding, cultural relevance, and the clarity of the topics. The students of the science club (sample of the research) received the final version of the questionnaire before and after the entire procedure, which can be found in Appendix A.

The Hake gain index (Hgain) was also used to quantify the pre- and post-change in students' awareness of the 4Cs skills because the sample size was relatively small and the level of the students varied in the different skills, as indicated by the data from the questionnaire before the TLS application. The change in group performance from a pre- to post-instruction assessment, by Hake gain index (g), is given in the relation (1).

$$g = \frac{(\text{average grade on the poste–test}) - (\text{average grade on the pre–test})}{100 - (\text{average grade on the pre–test})}, \quad (1)$$

The ratio in (1), also known as the normalised change (g), expresses the difference between average test scores as a fraction of the maximum difference conceivable between these scores. Hake [37] and Meltzer [38] utilised (1) to compare the relative effectiveness of various science course instructional strategies. Both researchers relied on the assumption that, when comparing two courses, "the course with the higher value of normalised change (g) is the more effective course" [39]. Hake [37] defines the levels of Hgain as follows: (a) "High-g" courses are those with Hgain > 0.7; (b) "Medium-g" courses are those with 0.7 > Hgain > 0.3; and (c) "Low-g" courses are those with Hgain < 0.3.

In addition, students were asked to answer paper-based open-ended questions after completing the questionnaire, and before and after the entire TLS. The open-ended questions contributed to a more in-depth investigation of students' awareness of the 4Cs skills and were based on the themes negotiated in the questionnaire. The open-ended questions can be found in Appendix B. Content analysis of the students' written open-ended answers was conducted by two authors and researchers of this paper. Firstly, the two researchers classified students' pre- and post- responses independently. Next, in order to resolve any differences between the two researchers, they discussed the data throughout the case until inter-rater reliability (0.9) was achieved.

### *3.3. Digital Mobile Equipment Used in the Implementation*

During the interventions, both conventional laboratory equipment (springs, weights, bases, clamps, etc.) and mobile technology (Smart Cart and Force Acceleration Sensors by PASCO and their related SPARKvue software installed on school tablets) were deployed. The Smart cart is a device with built-in sensors for tracking force, position, velocity, acceleration along three axes, and rotational velocity along three axes. The Wireless Force Acceleration Sensor is a device that simultaneously measures force, acceleration, and rotating velocity. These devices are paired with the PC suite software PASCO SPARKvue through Bluetooth for data logging on tablets/smartphones/laptops [40]. The collected data are visualised in a variety of formats (graphs, tables, numeric indicators, etc). An important aspect of SPARKvue suite is the capability to share the screen of the sensor-connected tablet with other students' tablets over the Internet (shared session). All students can save the experiment to their tablets and analyse the results after the experiment has been concluded. In our TLS, the sensor's dynamometer was utilised; the SPARKvue application took advantage of the sensor's ability to display measurement values on the screen and to provide these data in tabular and graphical formats.

Students' smartphones were also utilised for both student-to-student and student-to-teacher communication. This was accomplished by forming a Viber group, which provided students with an extra channel for information, communication, and cooperation via their smartphones, also outside the classroom environment.

### 3.4. Design and Implementation of the TLS

The study of factors that affect the development of teaching and learning sequences (TLSs) has looked at various aspects, including students' conceptions, epistemological assumptions, features of the educational context, etc. [41,42]. In this study, the focus is on the role of a specific teaching approach, which is aimed at increasing students' awareness of 4Cs skills. Therefore, a TLS was designed and implemented to determine the extent to which it contributes to students' awareness of 4Cs skills. Students worked in groups of three or four using mobile digital devices in an inquiry-based learning context (m-IBL) and were supported by worksheets; the worksheets were designed by considering their limited experience with group work, experimentation, inquiry-based learning, and mobile learning applications.

The three topics of the TLS, Hooke's Law, Vertical Spring Oscillation, and the Simple Pendulum, used the same inquiry-based learning framework according to Pedaste et al. [43]. Specifically, each topic was completed in four sessions (the duration of each session was two didactic hours), within a period of 12 weeks. Figure 1 depicts the phases of inquiry-based learning included in each topic through sessions.

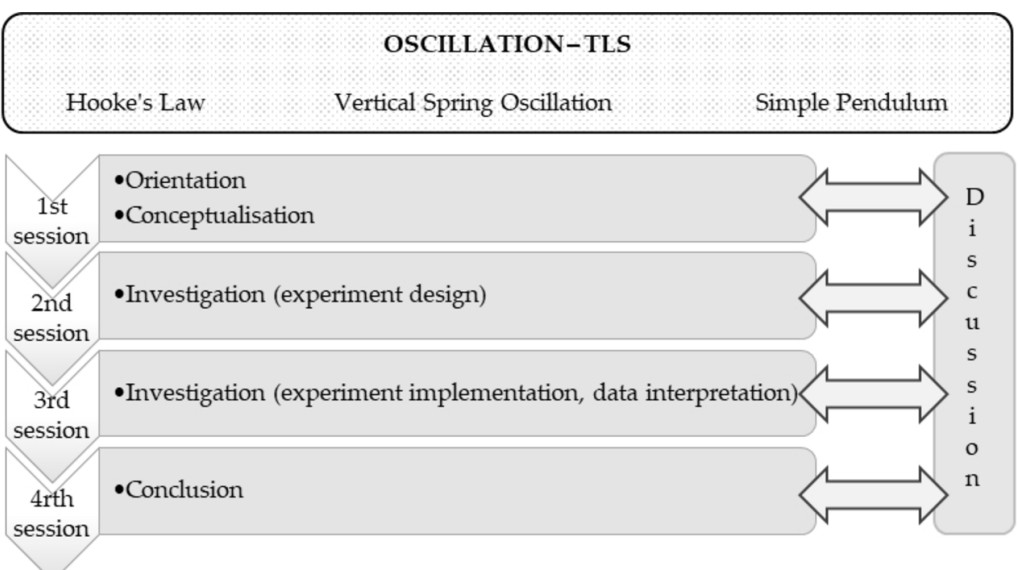

**Figure 1.** Structure of the phases of inquiry through sessions on each topic.

As seen in Figure 1, the first session corresponds to the Orientation and Conceptualisation phases of the Pedaste scheme, the next two sessions to the Investigation phase, and the final session to the Conclusion phase. The Discussion phase took place throughout the whole procedure. In the entire inquiry process, the students were oriented through a story based on everyday life, developed questions, and formulated hypotheses about the probable answers, designed and conducted experiments to test their hypotheses, analysed and evaluated the data, and drew conclusions. In addition, they evaluated the experimental procedure and communicated and discussed their findings in class. They also completed a Reflection Report at home during the entirety of the lab session.

The first session on each topic included the Orientation and Conceptualisation phases. In the Orientation phase, the students faced a fictionalized tale from their everyday lives to catch their interest and address a learning challenge. A representative extract of a fictionalized tale in the Orientation phase about the oscillation is as follows: "*John has not*

*decided whether he likes a mountain bike or a city bike, with the difference between the two being the spring stiffness of the suspension. John recognised the "stiffness" of a spring since he was taught Hooke's Law at school. However, he cannot correlate it with the bicycle's movement. At the shop, he was informed that the springs oscillate over road bumps, and he recalled that when he hanged a weight on a spring, it would move "up and down." So, he chose to analyse the oscillation of a vertical spring. On the occasion of the acquisition of the bicycle, Giannis inquired what elements impact on the oscillation period.".* In the Conceptualisation phase, students were prompted to generate testable questions and hypotheses on the presented problem and the fictitious scenario. The second session's purpose was to examine the fictional scenario. Students were asked to select lab equipment from the school's lab and design an experiment to test their hypotheses. During the third session, in each topic, the students experimentally verified the phenomenon. They conducted research, made observations, manipulated variables, and analysed findings. The students were introduced to the control-of-variables strategy by changing only one variable while keeping the others constant. Finally, the students used their tablets to record and analyse the data, and each group presented its results and made comparisons in front of the class. The fourth and last session, in each topic, comprised a review of the whole procedure followed by the students. Specifically, each student group received feedback and criticism from their classmates. The procedure was concluded by formulating the conclusions and comparing the findings to the initial hypotheses. Although reflection occurred throughout the whole procedure, students reflected on the entire inquiry-based learning experience during this final phase.

## 4. Results

The results on the students' awareness of 4CS skills were derived from the analysis of the students' written answers to the 3-point Likert questionnaire and their written answers to the open-ended questions, before and after experiencing mobile-technology-supported inquiry-based activities.

### 4.1. Evolution of Students' Awareness of 4Cs Skills

Figure 2 depicts the number of students who agree (value: 3, in the 3-point Likert scale) with each question of the questionnaire, before and after the TLS (dimensions: Collaboration—CL, Communication—CO, Critical thinking—CT, Problem solving—PS, and Creativity—CA). The data of Figure 2 are in Appendix C.

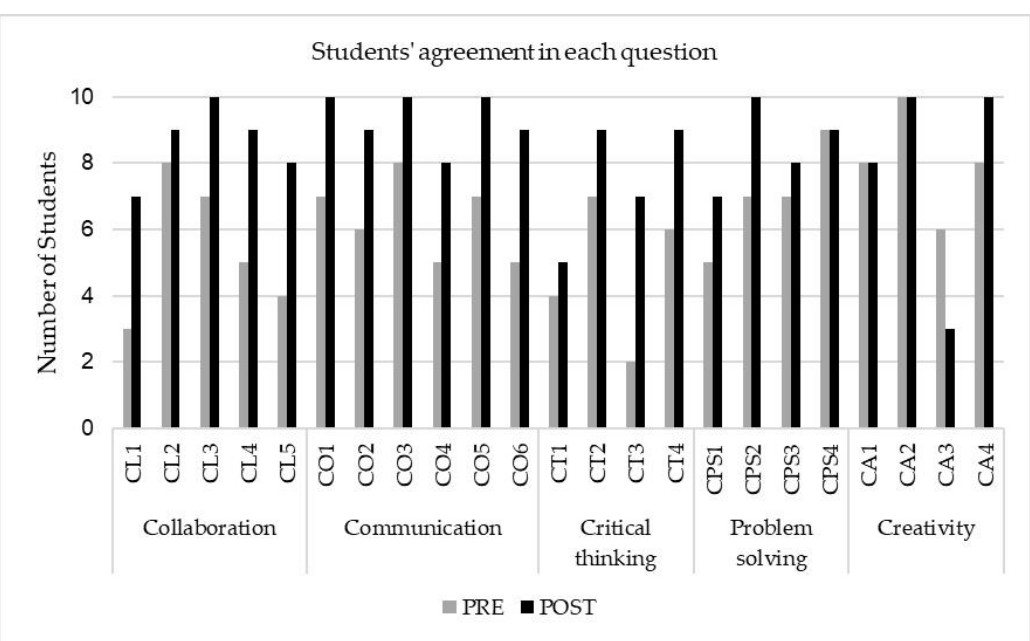

**Figure 2.** The number of students who agree with each question of the questionnaire.

As demonstrated in the graph, there is an improvement in students' agreement about almost all the questions of the questionnaire. Figure 2 also depicts that after the intervention, fewer students agree with one question (CA3) compared to the pre-test. The analysis of students' open-ended written answers, which are presented below in the paper, revealed the students' misunderstanding of question CA3, thus we did not include it in the analysis of the evolution of the students' awareness of 4Cs skills that follow.

Figure 3 displays the evolution of the students' awareness across the questionnaire's dimensions of Collaboration (CL), Communication (CO), Critical thinking (CT), Problem solving (PS), and Creativity (CA). As can be seen, the students' awareness of all skills improved. The data of Figure 3 are in Appendix C.

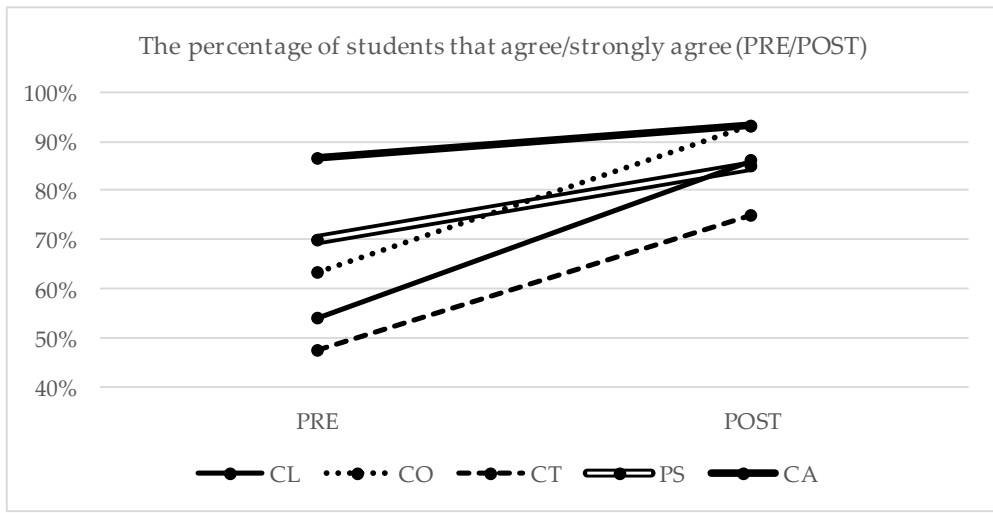

**Figure 3.** Evolution of the students' awareness across the questionnaire's dimensions.

Since the sample was very small (ten students) and the initial level of the students was not the same in the different skills, a clearer picture of their development can also be provided by the Hake increment, as seen in Figure 4. The data of Figure 4 are in Appendix C.

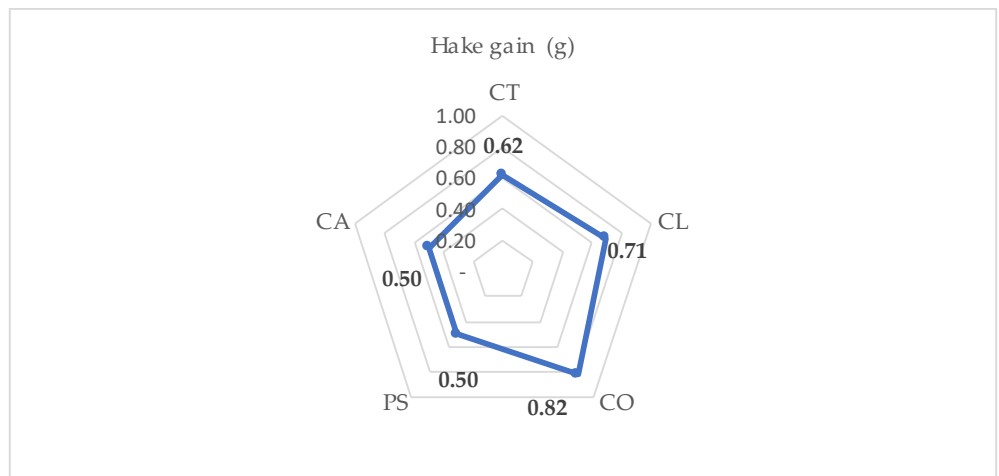

**Figure 4.** The Hake gain per skill.

As can be seen, the Collaboration and Communication skills present "High-g" (Hgain > 0.7), which means that there is a high difference between pre- and post- students' answers and the TLS could be characterized as highly effective. The other skills present Medium-g (0.7 > Hgain > 0.3), which means that there is a less but considerable difference between

pre- and post- students' answers and the TLS has also a positive impact on the students' awareness. Specifically, the Hake gain index of Critical thinking is 0.62, while the lowest Hake gain index (0.5) is observed in Problem solving and Creativity.

### 4.2. Students' Perceptions on 4Cs Skills Evaluated from Written Open-Ended Answers

The analysis of students' written answers to the open-ended questions contributed to the in-depth investigation of their awareness of the 4Cs skills. By looking for units of meaning in the answers of the students, we identified, with content analysis, characteristics which then were combined into codes and categories and were presented in each skill.

### 4.2.1. Collaboration

Students were asked to give written answers to three open-ended questions related to the Collaboration skill, along with the pre- and post-questionnaires, to illuminate their awareness in depth. The questions were: "What does it mean to you that team members collaborate effectively? Can you provide me with additional information about that?", "Do you feel you can do your assignment effectively in a group? Why?", and "How important do you find collaboration to be? Is it necessary, or can you accomplish your assignment on your own?". The students provided numerous and varied responses, which were classified and sorted into five categories, namely, Acknowledging group members' abilities through teamwork, Accomplishing the learning objectives, Promoting community, Sharing of authority, and Developing assertiveness, as depicted in Figure 5 (the data of Figure 5 are in Appendix C). Prior to the TLS, 12 answers were classified as the students' views for collaboration; after, the number of answers was increased to 26.

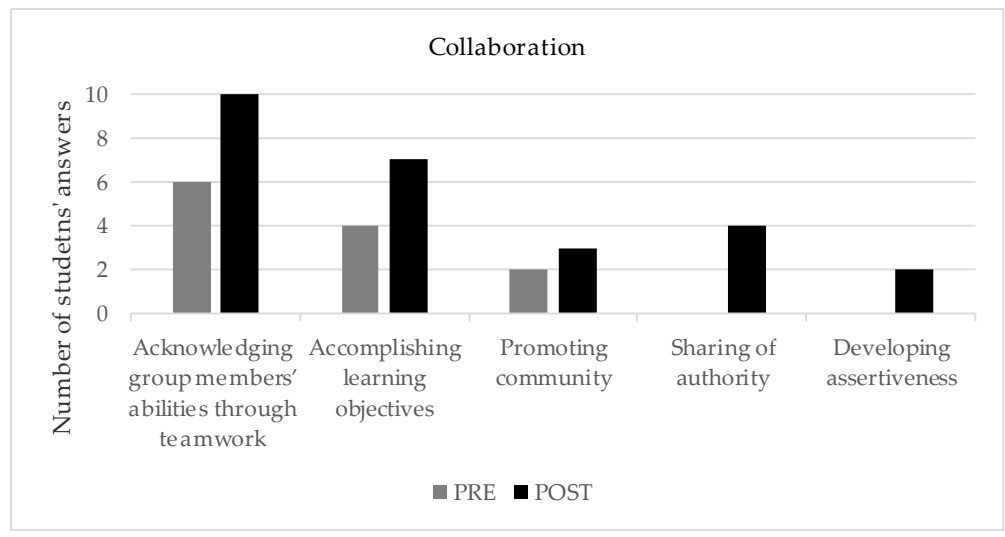

**Figure 5.** Categorizing students' written awareness of Collaboration.

Before the TLS, six out of ten students referred to "Acknowledging group members' abilities through teamwork" as part of Collaboration skills, while after the TLS, all the students mentioned this characteristic. Students' representative comments are as follows: S7 "*In my opinion, for successful cooperation, a group should be comprised of members with various personalities and skills. Teamwork has proven to be one of the most efficient strategies of mutual support*" and S9 "*Successful cooperation between the members of a group means the intersection of opinions with the aim of expressing all opinions for a perfect work!*". Four out of ten students mentioned the "Accomplishing learning objectives" as a benefit of the Collaboration before the TLS, while seven out of ten students referred to it at the end of the TLS. Representative students' answers which highlight the effectiveness of collaboration are: S1 "*Collaboration for me is very important because it is the best way to do a task. I can do it alone, but I believe that if I cooperate with other people, the work can be done more quickly, pleasantly, and efficiently*"

and S3 "*I realized that collective effort and cooperation among the members of a team leads to faster completion of the goal and reduces the chances of error*". It is worth mentioning the few students (two before the TLS and three after the TLS) who referred to the "Promoting community" gained through the collaboration instead of individualism, as they recognized an upper-level epistemological goal related to Collaboration skill. Indicatively, the student's S5 answer is: "*I believe it is crucial to think collaboratively. I believe that my classmates help me, but I can also contribute to the collaborative effort, which benefits us all.*".

The students also mentioned the "Sharing of authority" and "Developing assertiveness" related to Collaboration skills, after the TLS, which were not present in the pre-questionnaire. S2 stated that "*I believe that we should divide the tasks according to everyone's capabilities, so no one is held back, on the contrary, the sharing helps its members*"; this answer is indicative. Moreover, before the TLS, some negative comments regarding collaboration were recorded, while after the TLS, the students' comments on the same topic were only positive. Indicatively, before the TLS, student S7 had defended his disagreement with the collaboration, as follows: "*I believe I am unable to complete my group assignments efficiently due to the possibility of disagreements among individuals with various points of view.*" Additionally, indicative negative students' answers before the TLS include S8 stating that "*Collaboration is not effective as there are disagreements between the members of the group*" or S10 stating that "*By working individually you come out to learning objectives faster than working in a team*".

### 4.2.2. Communication

To highlight students' awareness of Communication skills, three open-ended questions were given to them, along with the pre- and post-questionnaires. The questions are as follows: "How do you try to make the other person feel important?", "Can you give an example on how you think about the other person's feelings when you are talking?" and "What does the term communicate with others mean to you?". Students provided numerous and varied responses, which were classified and sorted into five categories, namely, Participating in dialogue for expressing thoughts and ideas, Encouraging peers' effort, Communicating to attain certain goals, Developing empathy or respect for peers' personality, and Developing abilities of active listening, as depicted in Figure 6 (the data of Figure 6 are in Appendix C). Prior to the TLS, 16 answers were classified as students' views for communication, while after, the number of answers was increased to 24.

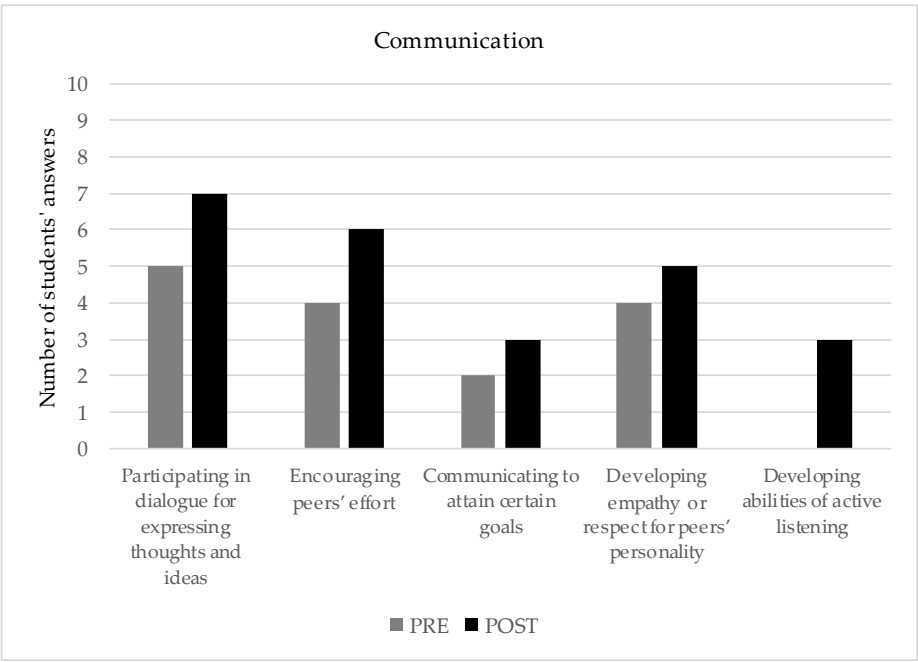

**Figure 6.** Categorizing students' written awareness of Communication.

As shown in Figure 6, students' awareness of communication skills has changed. Specifically, five out of ten students prior to the TLS, and seven out of ten students following the TLS, recognized "Participating in dialogue for expressing thoughts and ideas" as a characteristic of Communication skills. Representative students' answers are as follows: S2 *"For me, communicating with others means discussing and exchanging ideas so that we can learn from each other"* or S6 *"Communicating entails being able to speak openly, exchange perspectives, and not feel very uncomfortable around other members"*. The fact that the students were asked in each session, throughout the TLS, to present their experimental results to the class, may have contributed to the development of the students' awareness of this specific characteristic.

The evolution of students' awareness of "Encouraging peers' effort" and "Developing empathy or respect for peers' personality" between their pre- and post- answers is seen also in Figure 6. The teamwork during the experimentation in the sessions of the TLS and the continuous communication, even outside the classroom, through the Viber platform on students' mobiles, may have contributed to students' attempts to encourage and praise their peers, respect their viewpoints, and engage with them equally. Specifically, prior to the TLS, student S9 remarked: *"The other will feel important when he successfully completes his share of the work"*, without identifying how he would make his classmate feel important. The same student, following the TLS, commented: *"The other will feel significant when the rest of us recognise his work's effort."* Moreover, prior to the TLS, four students were able to give examples of thinking about peers' feelings by stating indicatively: S1 *"I answered I am not able to think about how the other person feels. So, I cannot give an example"*. After the TLS, various students' answers indicating their empathy and respect for peers' personalities are as follows: S5 *"Before expressing my perspective on a topic, I consider whether my words would offend or anger the listener"* or S8 *"Perhaps he will be offended if I say anything that makes him feel awful. For instance, if I tell him that his proposal is bad, he may not provide further suggestions and feel ineffective"*, while three students referred to the "Developing abilities of active listening", i.e., recognizing an epistemological goal gained and related it to Communication skills. Indicatively, S4 student wrote: *"When I communicate, I exchange information, ideas, knowledge, share concerns, listen carefully and take everyone's different opinions into account"*. It is worthwhile mentioning that no students mentioned the "Developing abilities of active listening" prior to the TLS, as shown in Figure 6, with the absence of pre-bar in this category.

Finally, less progress in recognizing goal achievement through communication skills is observed by students. Only two students prior to the TLS and three students after the TLS referred to this feature, answering indicatively: S10 *"When I speak with others, I am able to get along with them and accomplish my objectives"*. Goal achievement is more closely related to other skills, such as Collaboration and Creativity, as shown in the following results.

### 4.2.3. Critical Thinking and Problem Solving

The open-ended questions given to students (S1–S10) before and after the TLS, in order to investigate in-depth their awareness of Critical thinking and problem-solving skills, are: "If you make pauses, do you do so naturally or purposefully, i.e., do you find it essential to do so?", "What does it mean to you alternative solutions?", and "What makes you a strong problem solver?". Students' answers were classified and sorted by three characteristics of Critical thinking and problem-solving skills, namely, reflecting/monitoring, evaluating processes of thought, proposing alternative and viable solutions, and detecting and understanding the problem (Figure 7, the data of Figure 7 are in Appendix C). Prior to the TLS, 12 answers were classified as students' views for Critical thinking and problem solving, while after, the number of answers was increased to 26.

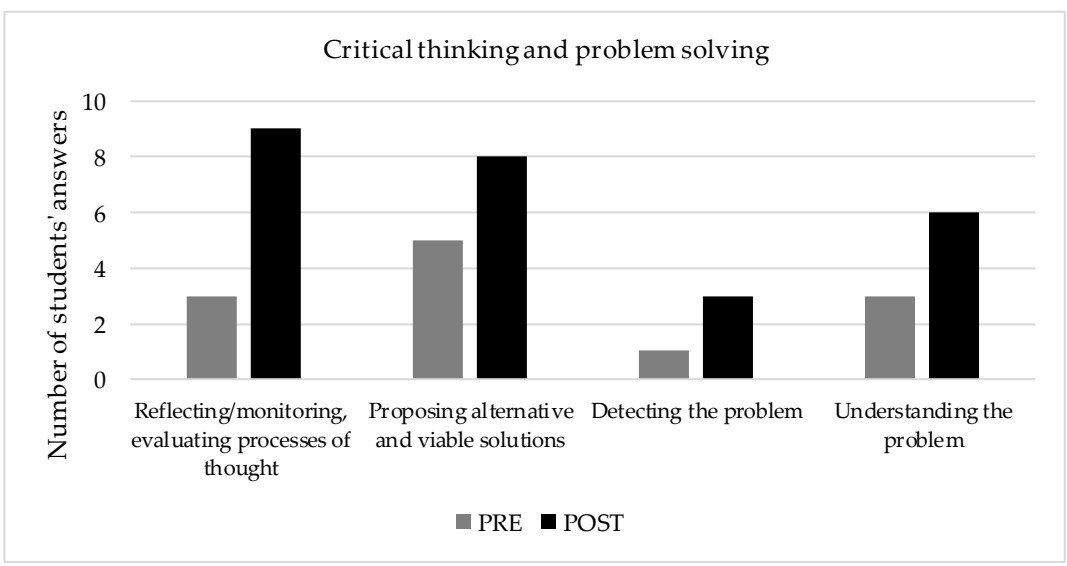

**Figure 7.** Categorizing students' written awareness of Critical thinking and problem solving.

As seen in Figure 7, prior to the TLS, three out of ten students indicated that they paused to reflect during the experiment, referring to the characteristic of "reflecting and evaluating thought processes" as part of Critical thinking skills. Following the TLS, nine out of ten students stated that they either intentionally or unintentionally paused to analyse and reflect on the procedure. Representative answers of students who stated that they make pauses, are: S1 "*Occasionally, I take a stop to evaluate if what I'm doing is correct or incorrect. This occurs naturally*" and S6 "*I pause spontaneously to think if I'm going the right way*".

The students were also asked to explain what the concept of "alternative solutions" meant to them. Figure 7 depicts that prior to the TLS, five out of ten students referred to different possible solutions to a problem, while after the TLS, eight out of ten students recognized the proposing of alternative and viable solutions as a feature of Critical thinking and problem-solving skills. Indicative students' answers are: S5 "*For me, alternative solutions entail considering all possible answers to an issue while simultaneously validating my results*", or S7 "*It signifies an alternative approach or seeing the issue from a different perspective*". It is worth noticing the parameter of a solution's efficacy besides the various approaches to solve a problem, as mentioned by some students after the TLS.

Four students before the TLS and nine students after the TLS included the "detecting and understanding the problem" in Critical thinking and problem-solving skills. Indicative students' answers to the question "What makes you a strong problem solver?" after the TLS are: S2 "*I am able to tackle problems because I thoroughly study, comprehend, and then effectively solve them*" or S9 "*My ability to utilise prior knowledge acquired from addressing earlier challenges enables me to solve the problems I meet*". Moreover, after the TLS, several students answered that the reason why they make pauses is to understand the problem they deal with. Representative students' answers are as follows: S4 "*I intentionally pause to determine whether I comprehend the actual issue*", or S5 "*If I do pause, I do it intentionally to comprehend the subject I am researching*". It is worth also noting the fact that students emphasized a variety of emotional or psychological traits, including "*calmness*", "*tenacity*", "*effort*", and "*patience and perseverance*". Before the TLS, indicative students' answers are: S2 "*What makes me good at solving problems is persistence, patience and good will*" or S9 "*I can solve problems because I put effort into it*". The continuous inquiry processes with analysis and evaluation of the data, in which the students were involved, may have strengthened the students' awareness of this skill.

#### 4.2.4. Creativity

The open-ended questions posed to students to explore in depth their awareness on creativity skills are: "Why do you like to try something new? What does this offer you?", "Do you like to do something by yourself? Why?", and "Why do you like to propose new ideas?". Students' responses were classified and sorted into three categories, namely, producing innovation (tangible or intangible), producing something beneficial in a social context, and gaining pleasure form producing innovation, as visualized in Figure 8 (the data of Figure 8 are in Appendix C). Prior to the TLS, 19 answers were classified as students' views for Creativity, while after, the number of answers was increased to 24.

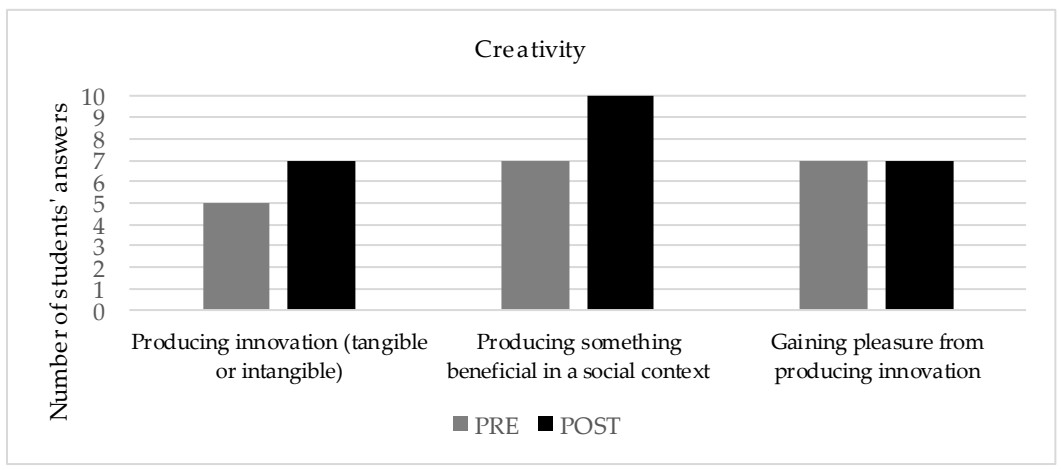

**Figure 8.** Categorizing students' written awareness of Creativity.

As seen in Figure 8, there is progress between pre- and post- answers related to "Producing innovation" and "Producing something beneficial in a social context". Moreover, Figure 8 shows that after the TLS, all the students referred to the "production of something that is beneficial in a social context". Representative students' answers are as follows: S1 "*I enjoy giving ideas because, when I do so, I usually assist others*" or S4 "*Contributing ideas makes me feel valuable and essential to my team*". This finding is not surprising, as students were asked to work in groups throughout the TLS, and the sense of contributing to the group was very strong.

The high level of students' answers to the "Gaining pleasure from producing innovation" was a feature related to Creativity skills, both before and after the TLS. Students' written answers, such as S3 stating that "*I enjoy sharing my perspective with others and giving them fresh ideas.*" or S6 "*When some of my ideas become valuable and I contribute to the solution of an issue, I experience a sense of satisfaction*", are representative. It is noteworthy to mention that six out of seven students who gave their consent both prior to and following the TLS were unchanged.

Finally, as we have already mentioned in Section 4.1, after the TLS, fewer students agreed with the question "CA3: I like to do things by myself" compared to the pre-test. The students' written answers to the open-ended questions conducted after the TLS were analysed to determine the cause of this negative attitude shift. Students were asked, "Why (or why not) do you enjoy doing things by yourself?". Some students reported, both before and after the TLS, that they love working alone because it makes it easier for them to concentrate on the topic, because doing anything alone helps them develop their abilities, or because it enables them to work independently. Representative answers are as follows: S1 "*I enjoy doing things by myself because this teaches me to solve difficulties independently, with tenacity and patience.*" Or S9 "*In this way, I learn myself better*". However, after the TLS, it appears that the positive aspects of teamwork that the students experienced had a detrimental impact on their attitude toward "doing something by myself". Representative students' answers are: S5 "*I no longer prefer working alone because after experiencing the*

*advantages of teamwork, I've realised that solitary effort is not as fruitful"*, S7 *"I despise working alone because I feel it provides fewer rewards than cooperation and collaboration"*, or *"When completing a group task with others, I feel more confident than when working alone"*.

## 5. Discussion

As essential components of inquiry-based learning, Collaboration, Communication, Critical thinking and problem solving, and Creativity skills were fostered throughout all phases of each topic in the TLS. Specifically, Collaboration skills were enhanced among the students as they were members of groups and were working together in all the inquiry phases. The whole procedure prompted students to develop Communication skills, as students were asked to present the experimental design, the analysis of the data, and the conclusions to their classmates and take part in discussions during each phase of the inquiry process. In addition, Critical thinking and problem solving and Creativity skills were predominantly cultivated during the formulation of the hypotheses while students designed the experiment to test their ideas and analysed the experimental data to come to conclusions.

The mIBL utilised in the TLS facilitated the inquiry process and contributed to the students' awareness of the 4Cs skills. In particular, transmitting the measurements recorded by the wireless sensors via Bluetooth to the students' tablets fostered both collaboration among the students and their critical thinking, as they had to communicate with each other to critically evaluate and interpret the data. In addition, the graphs and screenshots captured throughout the sessions stimulated fruitful debates, either through in-person interactions or through the Viber platform, to be used creatively in the dissemination of the research and its conclusions to the school as well as the scientific community. The analysis of the measurements (graphical or mathematical) allowed for the students to discuss and choose the most appropriate method for them, and in this way, to develop their creativity. Additionally, students' participation in their teacher's Viber group and the employment of their cell phones and the Viber platform for a different purpose than they were accustomed to, promoted cooperation and communication between them. From the results, it appears that the TLS contributed to improving the students' awareness of 21st century skills.

Regarding the first research question of the present study—to what extent mIBL affects students' awareness of the skills of Collaboration, Communication, Critical thinking and problem solving, and Creativity?—the Hake gain index showed that all the skills are High-g or Medium-g, which means that the TLS was effective. Specifically, the students' awareness of Collaboration skills underwent considerable change during the TLS. This conclusion is confirmed by the high Hake gain index (0.71). The students' awareness of Communication skills changed the most throughout the TLS—a conclusion supported by the high Hake gain index (0.82). Consequently, the TLS may be described as highly effective for these two skills. Regarding Critical thinking, after the TLS, there was an increase in students' agreement for all the questions of the questionnaire, except question 12, where student agreement was already too high to improve. Specifically, the Hake gain index of Critical thinking was 0.62. The lowest Hake gain index was observed in Problem solving and Creativity skills (0.5), although students referred to a variety of characteristics concerning these skills in their answers to the open-ended questions. Regarding the second research question of our study— what qualities/characteristics do students cite before and after the TLS in relation to Collaboration, Communication, Critical thinking and problem solving, and Creativity skills?—the results showed a variety of characteristics that students referred to in the open-ended questions. Specifically, more students acknowledged after the TLS than before it, the recognition of group members' abilities through cooperation, the achievement of learning objectives, the promotion of community, the exchange of authority, and the development of assertiveness as characteristics of Collaboration. The students' awareness of their Collaboration and Communication skills increased the greatest overall. The development in students' awareness of Communication skills was also demonstrated by their responses to open-ended questions. After the TLS, more students than before mentioned

the participation in dialogue for expressing thoughts and ideas, the encouragement of peers' effort, communication to attain certain goals, the development of empathy and respect for peers' personalities, and the development of active listening skills. The fact that all the TLS methods and practices (mobile learning, inquiry learning, and small group work) consistently encourage contact among learners may have had a significant impact on the range of Communication and Collaboration skill awareness.

The analysis of students' open-ended answers regarding Critical thinking and problem-solving skills confirmed the variety of characteristics students mention after the TLS. Specifically, nine out of ten students reported pausing throughout their work to reflect after the TLS, compared to just three before it; eight students reported proposing viable solutions after the TLS, compared to five before it; and nine students reported the detection and understanding of the problem after the TLS, compared to just four before it. It is also notable that after the TLS, students' written answers to open-ended questions contained elements related to cognitive processes, such as thinking, deciding, reasoning, and understanding the problem, which are essential for problem solving [44,45]. This is a noteworthy development in students' awareness since, as stated in the introduction, metacognition is a precursor or fundamental component of critical thinking. When engaging in critical thinking, students must evaluate their thought processes by determining if progress is being made toward an appropriate objective, assuring correctness, and determining how to allocate time and mental effort [17]. The improvement is even more significant when we consider that, while addressing a problem, students gain 21st century skills such as Collaboration, Communication, Critical thinking and problem solving, and Creativity, among others [46,47]. The findings of this study are in accordance with the conceptual frameworks that are proposed by several studies to describe the cognitive processes of problem solving, such as understanding the problem, constructing adequate representations of the problem, developing hypotheses, conducting experiments, and evaluating the solution [19,48], which are fundamental components of inquiry-based learning, which was implemented during the TLS.

As mentioned in the results section of this study, the students appeared to have already possessed a high level of creativity prior to the TLS, as indicated by their high percentage of agreement with the statement "I like to propose new ideas, no matter whether they are useful or not". This is further confirmed by their equally high rate of agreement with the question "I like to observe something I haven't seen before and understand it in detail", as creativity is about sensitivity in perceiving an issue [22], a process that begins with observation. Therefore, there was little room for development in these areas. However, an improvement in students' awareness of these skills was observed during the analysis of their answers to the open-ended questions. Specifically, seven out of ten students reported the production of innovation (tangible or intangible) after the TLS, compared to five before it, and all the students referred to the production of something beneficial in a social context after the TLS, compared to seven before it. At this point, it is also crucial to note that the misinterpretation of a questionnaire's item concerning creativity demonstrates the need to ensure that items in a questionnaire that analyse various dimensions of a topic do not lead to misunderstandings or conflicts between these dimensions.

## 6. Conclusions

Although there are many evidence-based insights into the relationship between IBL or m-learning and high-order thinking skills, there is a gap in the research about the impact of mIBL on the development of 21st century skills of students in Science Education. This study assesses the students' awareness of 4Cs skills after participating in mIBL. Additionally, the study investigates the qualities/characteristics students cite before and after the TLS in relation to these skills. The methodology, the design, and the implementation of a five-month mobile-technology-supported inquiry-based TLS are presented, together with the findings and conclusions drawn from qualitative research on the ninth-grade students' awareness of 4Cs skills. It appears that the mIBL TLS had a positive influence on all

skills. The results indicate that Collaboration and Communication skills had a greater impact than Critical Thinking and Problem solving, and Creativity. However, Hake gain was either high or medium across the board. The students' answers to the open-ended questions demonstrated an improvement in their skill awareness. After the TLS, students' responses were focused and targeted to each skill, as some students began to integrate scientific/research-based aspects in their responses, while prior to the TLS, students referred mainly to their general feelings (e.g., persistence, patience, composure).

Although the concepts of Collaboration, Communication, and Problem solving are almost clearly defined, it appears that the concept of Critical Thinking has several dimensions, and the concept of Creativity is quite vague. As a result, researchers use a wide variety of questionnaires for their research. Furthermore, a variety of cultural, socioeconomic, linguistic, and other variables influence the outcomes of studies carried out in different nations on the same issue; consequently, it is difficult to compare the results of different studies. Nonetheless, each study contributes to the students' understanding and cultivation of these skills and adds a tiny yet considerable amount to the study of higher-order thinking skills. The findings of our study represent a micro-level situation and can contribute to the diversification of research on the subject we are investigating. The limitation of the small sample leads us to plan a future implementation with a larger sample to confirm our conclusions. We also consider that more in-depth studies are necessary to provide insight into how students carry out the 4Cs skills and how to refine the TLS; this research is suggested to be carried out on a larger sample so that development will be substantially based on research evidence (the issue of iteration) [48].

**Author Contributions:** Conceptualization, M.K. and E.H.; methodology, M.K. and E.H.; software, M.K.; validation, E.P., A.M. and E.H.; formal analysis, M.K. and E.P.; investigation, M.K.; resources, M.K., E.P., A.M. and E.H.; writing—original draft preparation, M.K. and E.P.; writing—review and editing, A.M. and E.H.; supervision, A.M. and E.H. All authors have read and agreed to the published version of the manuscript.

**Funding:** This research received no external funding.

**Institutional Review Board Statement:** The study was conducted in accordance with the Declaration of Helsinki, and approved by the Institutional Review Board of 3rd Junior high school of Kavala (protocol code 494, 4 December 2021).

**Informed Consent Statement:** Informed consent was obtained from all subjects involved in the study.

**Data Availability Statement:** Data are not publicly available due to IRB guidelines, as well as consent and confidentiality agreements with participants. Any questions can be directed to the firstauthor.

**Conflicts of Interest:** The authors declare no conflict of interest.

## Appendix A. 4C1P Questionnaire

*Appendix A.1. Interaction with Learners*

Appendix A.1.1. Dimension 1: Collaboration (CL)

CL1 I believe our team can cooperate successfully when I conduct collaborative learning;

CL2 I try to provide useful and sufficient information when I conduct collaborative learning;

CL3 I have good communication with my team members when I conduct collaborative learning;

CL4 I can finish my work efficiently when I conduct collaborative learning;

CL5 Work is split based on our abilities when I conduct collaborative learning.

Appendix A.1.2. Dimension 2: Communication (CO)

CO1 I try to make the other person feel good;
CO2 I try to make the other person feel important;
CO3 I try to be warm when communicating with others;
CO4 While I'm talking I think about how the other person feels;
CO5 I am verbally and nonverbally supportive of other people;
CO6 I disclose at the same level that others disclose to me.

*Appendix A.2. Tendency of Higher-Order Thinking*

Appendix A.2.1. Dimension 1: Critical Thinking (CT)

CT1 I ask myself periodically if I am meeting my goals;
CT2 I consider several alternatives to a problem before I answer;
CT3 I find myself pausing regularly to check my comprehension;
CT4 I ask myself questions about how well I am doing once I finish a task.

Appendix A.2.2. Dimension 2: Problem Solving (PS)

PS1 When facing problems, I believe I have the ability to solve them;
PS2 I believe I can put effort into solving problems;
PS3 I can solve problems that I have met before;
PS4 I am willing to face problems and make an effort to solve them.

Appendix A.2.3. Dimension 3: Creativity (CA)

CA1 I like to observe something I haven't seen before and understand it in detail;
CA2 I like to try something new;
CA3 I like to do something by myself;
CA4 I like to propose new ideas, no matter they are useful or not.

**Appendix B. Open-Ended Questions**

| Skills | Open-Ended Questions |
|---|---|
| Collaboration | What does it mean to you that team members collaborate effectively? Can you provide me with additional information about that? |
| | Do you feel you can do your assignment effectively in a group? Why? |
| | How important do you find collaboration to be? Is it necessary, or can you accomplish your assignment on your own |
| Communication | How do you try to make the other person feel important? |
| | Can you give me an example of how you think about the other person's feelings when you are talking? |
| | What does the term communicate with others mean to you? |
| Critical thinking and problem solving | If you make pauses, do you do so naturally or purposefully, i.e., do you find it essential to do so? |
| | What does it mean to you alternative solutions? |
| | What makes you a strong problem solver? |
| Creativity | Why do you like to try something new? What does this offer you? |
| | Do you like to do something by yourself? Why? |
| | Why do you like to propose new ideas? |

## Appendix C. Tables with the Data of the Figures

**Table A1.** Data for Figure 2.

| | | Number of Students | |
| --- | --- | --- | --- |
| | | PRE | POST |
| **Collaboration** | CL1 | 3 | 7 |
| | CL2 | 8 | 9 |
| | CL3 | 7 | 10 |
| | CL4 | 5 | 9 |
| | CL5 | 4 | 8 |
| **Communication** | CO1 | 7 | 10 |
| | CO2 | 6 | 9 |
| | CO3 | 8 | 10 |
| | CO4 | 5 | 8 |
| | CO5 | 7 | 10 |
| | CO6 | 5 | 9 |
| **Critical thinking** | CT1 | 4 | 5 |
| | CT2 | 7 | 9 |
| | CT3 | 2 | 7 |
| | CT4 | 6 | 9 |
| **Problem solving** | CPS1 | 5 | 7 |
| | CPS2 | 7 | 10 |
| | CPS3 | 7 | 8 |
| | CPS4 | 9 | 9 |
| **Creativity** | CA1 | 8 | 8 |
| | CA2 | 10 | 10 |
| | CA3 | 6 | 3 |
| | CA4 | 8 | 10 |

**Table A2.** Data for Figure 3.

| | | Percentage of Students | |
| --- | --- | --- | --- |
| | | PRE | POST |
| **4Cs Skills** | CT | 48% | 75% |
| | CL | 54% | 86% |
| | CO | 63% | 93% |
| | PS | 70% | 85% |
| | CA | 87% | 93% |

**Table A3.** Data for Figure 4.

| | | Hake Gain (g) |
| --- | --- | --- |
| **4Cs Skills** | CT | 0.62 |
| | CL | 0.70 |
| | CO | 0.82 |
| | PS | 0.50 |
| | CA | 0.50 |

**Table A4.** Data for Figure 5.

| | | Number of Students | |
|---|---|---|---|
| | | PRE | POST |
| ollaboration | Acknowledging group members' abilities through teamwork | 6 | 10 |
| | Accomplishing learning objectives | 4 | 7 |
| | Promoting community | 2 | 3 |
| | Sharing of authority | 0 | 4 |
| | Developing assertiveness | 0 | 2 |

**Table A5.** Data for Figure 6.

| | | Number of Students | |
|---|---|---|---|
| | | PRE | POST |
| Communication | Participating in dialogue for expressing thoughts and ideas | 5 | 7 |
| | Encouraging peers' effort | 4 | 6 |
| | Communicating to attain certain goals | 2 | 3 |
| | Developing empathy and respect for peers' personality | 4 | 5 |
| | Developing abilities of active listening | 0 | 3 |

**Table A6.** Data for Figure 7.

| | | Number of Students | |
|---|---|---|---|
| | | PRE | POST |
| Critical thinking and problem solving | Reflecting/monitoring, evaluating processes of thought | 3 | 9 |
| | Proposing alternative and viable solutions | 5 | 8 |
| | Detecting the problem | 1 | 3 |
| | Understanding the problem | 3 | 6 |

**Table A7.** Data for Figure 8.

| | | Number of Students | |
|---|---|---|---|
| | | PRE | POST |
| Creativity | Producing innovation (tangible or intangible) | 5 | 7 |
| | Producing something beneficial in a social context | 7 | 10 |
| | Gaining pleasure from producing innovation | 7 | 7 |

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
