# Peer review of "Assessing Students’ Awareness of 4Cs Skills after Mobile-Technology-Supported Inquiry-Based Learning"

_sustainability, doi:10.3390/su15086725_

Round 1

Reviewer 1 Report

Overall, the manuscript has good merit, but there could be some consistencies and points of clarification throughout. Please see the itemized areas below:

1. Line 30, the space in "educators" should be removed.

2. 4Cs needs to be consistent throughout. Ex: Critical thinking and problem solving is first in the 2nd paragraph of the introduction, but appears last in the abstract and first paragraph of section 2.1. Also line 180-181, 4Cs are in a different order, also line 202.

3. Line 164, "obtaining information several nations" is unclear. This thought reads incomplete.

4. What is the digital literacy/prior mobile learning experience of 9th graders prior to the study?

5. Line 207, expand on what "great care" actually means. This term sounds very vague. 

6. The study's small sample size limits its power and generalizability. This should be accounted for in the Limitations section.  The qualitative quotes help to provide perspective.

7. The pre-bar for Developing abilities of active listening is missing from figure 6.

8. Future Implications for the Discussion could include an observational study on HOW students carry out creativity, communication, problem solving, etc.

9. Line 543 "problem solving" is missing after Critical Thinking.

10. The second paragraph of the Discussion sounds like a Methods section.

11. Line 578 gain index 0.62, reads as an incomplete sentence.

12. Line 583-584, again 4Cs are in different order.

13. Line 609, "C" should be lowercase.

14. The last paragraph of the Discussion reads as a baseline on creativity from qualitative data. I would like to see this for the other 3Cs as well.

15. Line 648-649, Given the psychological and emotional components as explanation of concepts, how could this aid in shaping teen social-emotional development?

16. Please make a clear connection on how the m-health tool facilitates 4C self awareness in the 9th grade sample.

Reviewer 2 Report

The article has an essential theme in the educational field. Part of the significant learning with the new methodologies. The contribution he makes is interesting, making a search for important literature within the subject. The main limitation is the sample, perhaps too small. It could be solved by indicating it in the conclusions, for future similar investigations. On the other hand, the figures do not include where they have been extracted from, it would be convenient to provide it, if it were self-made, it should be explicit. I believe that the article could be considered for publication if these two aspects are taken into account.

Reviewer 3 Report

The proposed article reveals great topicality for the academic debate and proposes innovation in the pedagogical methods of teaching-learning. It has a great contribution to the new thinking in Education and shows potential to be replicated in a more macro context.

Round 2

Reviewer 1 Report

This paper is much improved and should be considered for publication.